# Does Amputation Negatively Influence the Incidence of Depression in Diabetic Foot Patients? A Population-Based Nationwide Study

**Dong-Il Chun** [1,†], **Jahyung Kim** [1,†], **Eun Myeong Kang** [1], **Chi Young An** [1], **Tae-Hong Min** [1], **Sangyoung Kim** [2], **Jaeho Cho** [3], **Young Yi** [4] and **Sung Hun Won** [1,*]

1   Department of Orthopaedic Surgery, Soonchunhyang University Seoul Hospital, 59 Daesagwan-ro, Yongsan-gu, Seoul 04401, Korea; orthochun@gmail.com (D.-I.C.); hpsyndrome@naver.com (J.K.); 129741@schmc.ac.kr (E.M.K.); achi0106@naver.com (C.Y.A.); minth916@gmail.com (T.-H.M.)
2   SCH Biomedical Informatics Research Unit, Soonchunhyang University Seoul Hospital, 59 Daesagwan-ro, Yongsan-gu, Seoul 04401, Korea; kkimsy@naver.com
3   Department of Orthopaedic Surgery, Chuncheon Sacred Heart Hospital, Hallym University, 77, Sakju-ro, Chuncheon-si 24253, Korea; hohotoy@nate.com
4   Seoul Foot and Ankle Center, Department of Orthopaedic Surgery, Inje University Seoul Paik Hospital, 85, 2-ga, Jeo-dong, Jung-gu, Seoul 04551, Korea; 20vvin@naver.com
*   Correspondence: orthowon@gmail.com; Tel.: +82-10-709-9250; Fax: +82-2-710-3191
†   These authors contributed equally to this work.

**Abstract:** This study aimed to investigate the relationship between diabetic foot ulcer and depression based on treatment methods employed, as evaluated according to Medicare claims data provided by the Health Insurance Review and Assessment Service (HIRA). Data on diabetic foot patients from January 2011 to December 2016 were collected from the HIRA using codes for diabetic foot and depression disorder. The incidence of depression was analyzed based on patients' demographic variables, and comorbidities were assessed using the Charlson comorbidity index (CCI). The participants were divided into two groups based on the treatment method used: a limb-saving group and an amputation group. The 1-, 3-, and 5-year incidence rates of depression were 10.1%, 20.4%, and 29.5%, respectively, in the limb-saving group and 4.5%, 8.2%, and 11.5%, respectively, in the amputation group. Female sex, the CCI, and the use of limb-saving treatment methods were significant risk factors. It is plausible that depression in diabetic foot patients may be associated with frequent recurrence and chronicity rather than a single intense event. Our findings highlight the need for clinicians to consider the treatment period as a contributor to patient mood disorders when selecting the appropriate course of action in patients.

**Keywords:** diabetic foot; amputation; depression

## 1. Introduction

With the increase in the aging of the global population, the prevalence of diabetes mellitus is also expected to increase [1]. Moreover, the regulation of diabetes mellitus is challenging, requiring a significant duration of management and treatment. As a result, dealing with uncontrolled glucose levels may result in a substantial burden for patients, not only economically but also emotionally [2,3].

Diabetic foot ulcer is a commonly observed complication in diabetes mellitus patients and is associated with various amputation rates and life-threatening complications. Only two-thirds of all foot ulcers eventually heal and up to 28% may warrant lower extremity amputation [4–7]. The coexistence of mental disorders and physical illness has been highlighted in previous studies and it is known that 39.6% of patients with diabetic foot experience depression [8].

Depression is a mood disorder that includes several symptoms that alter the functionality of an individual [9]. Major depression is characterized by depressed mood, lack of interest in enjoyable activities, increase or decrease in appetite, insomnia or hypersomnia, slowing of movement, lack of energy, feelings of guilt or worthlessness, trouble concentrating, and suicidal thoughts or behaviors [10,11]. In addition, it may lead to social problems, such as familial separation and socio-economical loss [12–15]. However, it remains unclear whether depression in diabetic foot patients results from the treatment duration, the loss of a body part through amputation, or uncertainty pertaining to the future. In previous studies, there were reports of the psychological effects of diabetic foot ulcers, in terms of psychological symptoms such as depression, but reports about the effects of limb amputation were lacking [8,16–21]. In fact, to the best of our knowledge, no multi-year, longitudinal study has focused on the relationship between depression and amputation.

Accordingly, this study sought to determine the effects of amputation and chronic wounds in the development of depression and to analyze the associated risk factors in patients with diabetic foot, using Medicare claims data provided by the Health Insurance Review and Assessment Service (HIRA).

## 2. Materials and Methods

### 2.1. Study Design and Patients

The codes directly indicative of diabetic foot were established on 1 January 2011, following the implementation of the Korean Statistical Classification of Disease and Related Health Problems-6 system. Prior to 2010, the disease codes for diabetic foot gangrene and ulcer were separated from the disease code for diabetes. In the absence of appropriate documentation of the disease code for foot wounds, patients with diabetic foot were excluded. Therefore, the current study did not investigate data obtained before 2010, and only included data from 2011. Therefore, this study utilized data for 127,649 patients during the six-year period from 1 January 2011 to 31 December 2016, based on information obtained from the HIRA. Moreover, a one-year wash-out period was set for the determination of the incidence of diabetic foot. Accordingly, data pertaining to the incidence of newly diagnosed diabetic foot among 65,052 patients, from 2012, were included (Figure 1).

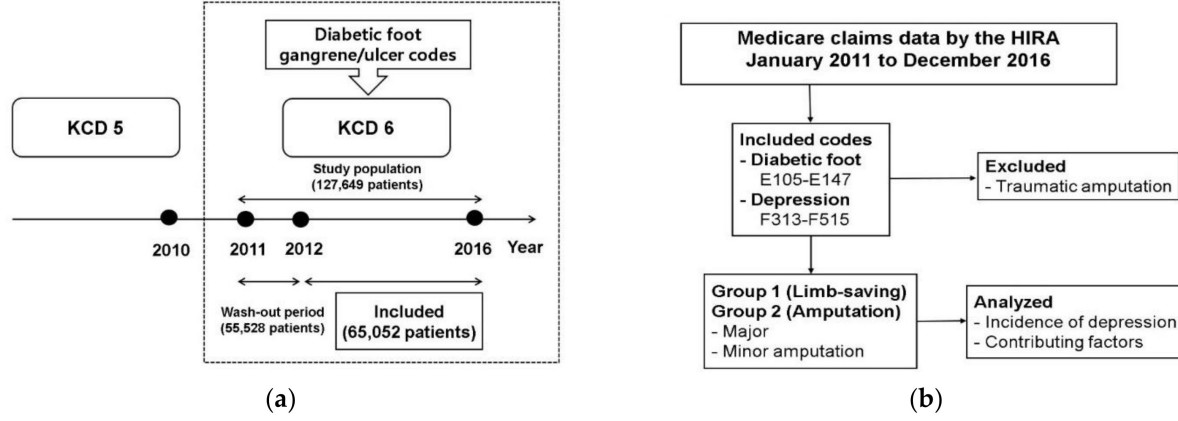

**Figure 1.** (**a**) Diagram showing the study period and included patients; (**b**) Flow diagram of the study. KCD = Korean Statistical Classification of Disease and Related Health Problems. HIRA = Health Insurance Review and Assessment Service.

The codes for diabetic foot and depressive disorder, and the treatment methods employed in our study, are summarized in the Appendices A and B. In terms of amputation types, those amputations involving the thigh (N0572) and tibia (N0573) were classified as major amputations, while those performed on the foot (N0574) and the phalanx (N0575) were classified as minor amputations (Table 1).

**Table 1.** Codes included: diagnosis and procedure.

| Diagnosis | Procedure |
|---|---|
| **Diabetic foot**<br>E1050, E1051, E1058, E1070, E1071, E1072, E1078,<br>E1150, E1151, E1158, E1170, E1171, E1172, E1178,<br>E1250, E1251, E1258, E1270, E1271, E1272, E1278,<br>E1350, E1351, E1358, E1370, E1371, E1372, E1378<br>E1450, E1451, E1458, E1470, E1471, E1472, E1478 | **Limb-saving**<br>Debridement (SC021-SC027)<br>Simple dressing (M0111)<br>Infectious wound dressing (M0121)<br>Suction drainage (M137) |
| **Depression**<br>F204, F251, F313, F314, F315, F320, F321, F322, F323,<br>F328, F329, F330, F331, F332, F333, F334, F338, F339,<br>F412, F920 | **Amputation**<br>Minor amputation<br>Amputation, thigh (N0572)<br>Amputation, tibia (N0573)<br>Major amputation<br>Amputation, foot (N0574)<br>Amputation, phalanx (N0575) |

Diabetic foot patients were categorized into two groups: a limb-saving group ($n$ = 65,052) and an amputation group ($n$ = 3793) (Figure 1).

In both groups, the incidence of depression was defined as the time from the start of the treatment for diabetic foot to the time of the diagnosis of depressive disorder. Kaplan-Meier curves were utilized for the visualization of the depression rate.

### 2.2. Statistical Analysis

Sex, age, and the number of comorbid diseases were included as variables. For the analysis of the number of comorbid diseases, the Charlson comorbidity index (CCI) was adopted [22]. The amputation group was further categorized into major and minor amputation sub-groups. Propensity score matching was performed on sex, age, and the CCI for balance between the groups, while Cox regression analyses were used for the comparison of the hazard ratios (HRs) between the variables. $p$-values of < 0.05 were considered statistically significant. All data in this study were analyzed using the SAS Enterprise Guide, ver. 6.1 M1 (SAS Institute Inc., Cary, NC, USA).

### 2.3. Ethics Statement

This retrospective study was approved by the Institutional Review Board of Soonchun-hyang University Hospital Seoul (SCHUH 2018-01-007). Informed consent was waived by the Board.

## 3. Results

### 3.1. Incidence of Depression

Overall, the postoperative 1-year incidence rates of depression were 10.1% and 4.5%, while the 3-year/5-year rates were 20.3%/8.2% and 29.5%/11.5% in the limb-saving and amputation groups, respectively. The incidence of depression was lower in the amputation group (Figure 2).

### 3.2. Contributing Factors

In the Cox regression analysis, performed for the identification of the depression-related risk factors, age did not show a statistically significant effect (HR, 1.004; 95% CI, 1.000–1.009; $p$ = 0.056). However, female sex was associated with a significantly higher risk of depression development than male sex (HR, 1.275; 95% CI, 1.117–1.451; $p$ < 0.001). In the analyses, which were performed based on the comorbid diseases, the risk of depression increased with the number of underlying diseases. The effect was statistically significant (HR, 1.125; 95% CI, 1.098–1.151; $p$ < 0.001). Of the treatment methods, the limb-saving method was associated with three times the risk of depression development compared to the risk associated with amputation (HR, 2.805; 95% CI, 2.478–3.183; $p$ < 0.001).

Finally, no significant difference was identified between the minor amputation and major amputation groups (HR, 1.036; 95% CI, 1.000–1.077; *p* = 0.690) (Table 2). The incidence of depression was lower in the amputation group.

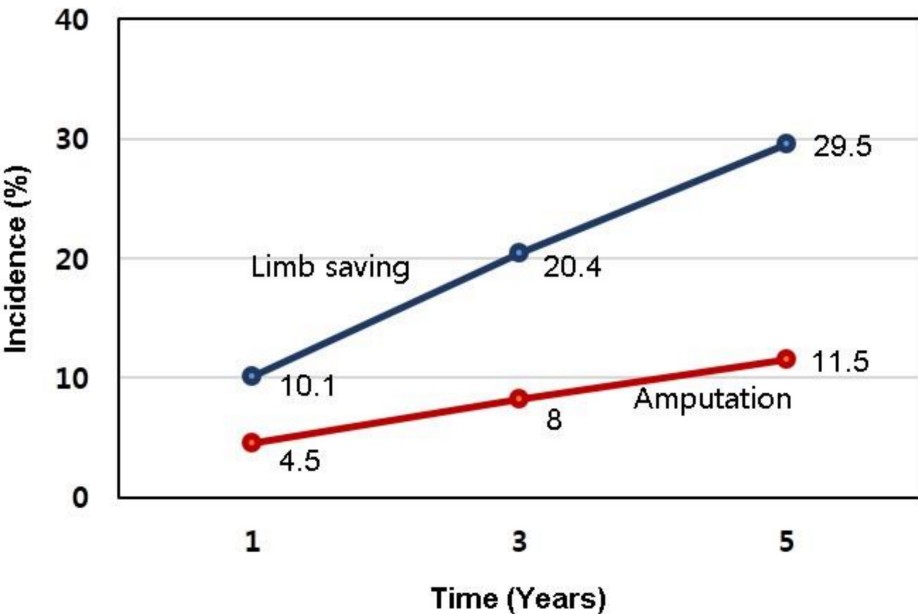

**Figure 2.** Comparison of the incidence of depression between the limb-saving group and amputation groups. Time (years) is defined as the duration from the initiation of the diabetic foot treatment.

**Table 2.** Cox regression analysis.

| Variables | HR | 95% CI | | *p*-Value |
|---|---|---|---|---|
| | | Lower | Upper | |
| Sex | | | | |
| Male | Reference | | | |
| Female | 1.275 | 1.117 | 1.451 | <0.001 |
| Age | 1.004 | 1.000 | 1.009 | 0.055 |
| CCI | 1.125 | 1.098 | 1.151 | <0.001 |
| Treatment method | | | | |
| Limb-saving | 2.805 | 2.478 | 3.183 | <0.001 |
| Amputation | Reference | | | |
| Amputation type | | | | |
| Minor | 1.036 | 1.000 | 1.077 | 0.69 |
| Major | Reference | | | |

HR = hazard ratio, CI =cConfidence interval, CCI = Charlson comorbidity index.

## 4. Discussion

In the current study, the incidence of depression increased in association with the treatment period among patients with diabetic foot. Limb-saving treatment was associated with a higher depression rate than amputation. With regard to risk factors contributing to the risk of depression development, sex, number of comorbid diseases, and chronic limb-saving procedure were found to be significant factors.

Nonsurgical limb-saving treatment for diabetic foot ulcers involves interventions for local pathologic factors, including debridement, pressure relief, antibiotic treatment for infection, and revascularization. Such procedures, while providing the benefits of limb-saving and mobility maintenance, are time-consuming, leading patients to worry about their future [3]. Monami et al. [23] reported higher depression scores in association with

delayed diabetic foot ulcer healing. Another study reported progression in the degree of mood impairment in patients among whom the ulcers failed to heal despite lengthy treatment periods [24]. Conversely, amputation, which may have a positive effect in shortening a patient's treatment period, may have a preventive effect on his/her mood compared to a limb-saving procedure, as it provides a definite result. A previous study reported that minor amputation does not negatively affect patients' health-related quality of life, compared to conservative treatment in patients with diabetic foot ulcers [25]. Therefore, in terms of depression, surgeons' efforts that are driven towards retaining patients' body parts may not positively influence their mental well-being, owing to treatment period prolongations and uncertainties.

In this study, women with diabetic foot were more susceptible to depression than men, which is consistent with the results of previous studies [8,26]. Ahmed et al. [8] indicated that women are more aware of their body parts, and therefore tend to utilize health services more often than men. In addition, women tend to be the caregivers for both themselves and their family members in cases of chronic diseases [27]. They are also more subject to sex-specific conditions, such as pregnancy, menstrual cycle changes, and postpartum stress [26]. These factors may have a strong effect on a woman's feeling of endangerment in terms of the complications associated with foot ulcer.

Although the current study did not identify an association between depression and age, most previous studies reported that patients with depression tend to be younger [18,28]. From the socioeconomic point of view, younger people who tend to be the breadwinners for themselves and their family may perceive chronic foot ulcer in a more serious light than do older people. Additionally, owing to the widespread individualism in modern society, characterized by restricted lives and a lack of social support, younger diabetic foot ulcer patients are more susceptible to the development of a depressed mood [16].

In line with other studies, we found that depression among diabetes patients was directly related to the coexistence of other chronic diseases [17,29]. The degree of chronicity of the coexisting diseases may have an effect on a patient's physical condition, leading to delayed wound-healing and precarious prognoses. In addition, diabetes patients with several other complications, such as retinopathy, nephropathy, and vasculopathy, usually experience difficulties in adhering to a diet, exercising, and complying with prescribed medications [26]. Lack of compliance may lead to the development and progression of microvascular and macrovascular complications, eventually resulting in foot ulcer deterioration [30]. These series of events associated with the presence of chronic comorbidities may ultimately result in a depressed mood.

This study had some limitations. First, the codes for diabetic foot and depression disorder were diverse, indistinct, and sometimes lacking. Before 2010, there was no defined disease code for diabetic foot; thus, data on diabetic foot from 2007 to 2010 could not be utilized. This limitation is related to the use of the Medicare claims data provided by the HIRA. In the future, it will be necessary to agree on codes that are more clearly and uniformly indicative of diabetic foot and depression disorder. Second, in the evaluation of the incidence of chronic diseases, such as diabetes, a wash-out period of at least 2 years should be employed [31]. However, we used a 1-year wash-out period, due to data insufficiency. This might have influenced the results of this study. Third, there is a further limitation in that data after 2016 are not included. Therefore, if access to HIRA data after 2016 is available, further analysis would benefit from this information. Lastly, a patient's depression can be caused by other factors, such as a second or third amputation, antiglycemic treatment (e.g., change from pills to injections), the development of neuropathic pain, or other complications. Therefore, future studies should be considered that include these factors.

Our study has several strengths. To the best of our knowledge, it is the first multi-year, large-scale study focusing on the incidence of depression in patients with diabetic foot in terms of the treatment methods employed. Furthermore, this study attempted to clarify the distinct sources of evidence on depressed mood among patients with diabetic foot ulcers.

## 5. Conclusions

The prevalence of depression disorder was found to be high among patients with diabetic foot, especially among female patients, those with coexisting physical illness, and those treated with limb-saving procedures. It is plausible that repetitive minor treatments, which are associated with a greater degree of uncertainty and chronicity, may have a stronger effect than a single major event such as an amputation. Therefore, health professionals need to follow clinical practice guidelines in selecting optimal treatment methods for diabetic ulcer. Limb amputation is only necessary for the unsalvageable diabetic foot, regardless of patients' depressive symptoms. Health professionals should pay more attention to the mental health of their patients in both the amputation and limb-saving groups. It is recommended that clinicians pay a higher level of attention to depressive mood changes that occur due to treatment period prolongations in the selection of optimal treatment methods for diabetic ulcer.

**Author Contributions:** D.-I.C. designed and supervised the study; J.K. wrote the initial draft of the manuscript and analyzed the data; E.M.K., C.Y.A. and T.-H.M. reviewed and edited the draft manuscript; S.K. performed the statistical analyses; J.C. and Y.Y. critically revised the manuscript; S.H.W. served as the guarantor for this work, had full access to all the data used in the study, and takes responsibility for the integrity of the data and the accuracy of the data analysis. All authors have read and agreed to the published version of the manuscript.

**Funding:** This work was supported by the Soonchunhyang University Research Fund (1021-0001). This work was supported by the National Research Foundation of Korea(NRF) grant funded by the Korea government(MIST) (No. 2020R1I1A3071875).

**Institutional Review Board Statement:** The study was conducted according to the guidelines of the Declaration of Helsinki, and approved by the Institutional Review Board of Soonchunhyang University Hospital Seoul (SCHUH 2018-01-007).

**Informed Consent Statement:** Not applicable.

**Data Availability Statement:** Supporting information about codes for diabetic foot, depressive disorder and the treatment methods is available at https://www.koicd.kr/.

**Conflicts of Interest:** The authors have no conflicts of interest to disclose.

## Appendix A. Diabetic Foot Codes

E1050-1058
Type 1 diabetes mellitus, with diabetic peripheral angiopathy
E1070-1078
Type 1 diabetes mellitus, with diabetic foot ulcer
E1150-1158
Type 2 diabetes mellitus, with diabetic peripheral angiopathy
E1170-1178
Type 2 diabetes mellitus, with diabetic foot ulcer
E1250-1278
Malnutrition-related diabetes mellitus, with diabetic peripheral angiopathy
E1350-E1358
Other specified diabetes mellitus, with diabetic peripheral angiopathy
E1370-1378
Other specified diabetes mellitus with, diabetic foot ulcer
E1450-1458
Unspecified diabetes mellitus, with diabetic peripheral angiopathy
E1470-1478
Unspecified diabetes mellitus, diabetic foot ulcer

## Appendix B. Depression Codes

| | |
|---|---|
| F204 | Post-schizophrenic depression |
| F251 | Schizoaffective disorder, depressive type |
| F313-315 | Bipolar affective disorder |
| F320-329 | Depressive episode |
| F330-339 | Recurrent depressive disorder |
| F412 | Mixed anxiety and depressive disorder |
| F920 | Depressive conduct disorder |

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
