# Peer review of "Does Amputation Negatively Influence the Incidence of Depression in Diabetic Foot Patients? A Population-Based Nationwide Study"

_applsci, doi:10.3390/app12031653_

Round 1

Reviewer 1 Report

The current study is an intriguing study on the incidence of depression in diabetic foot patients. However, it is an unexpected finding that depression is higher among saving-limb patients. An explanation could be partly because the amputation group is small compared to saving-limb. Another reason could be the kind of treatment between the two groups. I suppose that saving-limb patients had better glycemic control (much effort needed), more frequent visits to their doctors, and maybe a more appropriate treatment for all other comorbidities.

Therefore, the manuscript could be more interesting if the authors could provide data about glycemic control, HbA1c values, and the kind of treatment received.

I think that the methodology is appropriate. The manuscript is generally clearly written, and the discussion/conclusions are acceptable.

   Overall, the data could be interesting if the authors added the abovementioned data

Author Response

  • We authors are really thank the reviewer for the thoughtful advice on our work. We do agree on the point that other factors such as glycemic control can influence the depression rate between two groups. However since our study is based on Medicare claims data pro-vided by the Health Insurance Review and Assessment Service (HIRA), only codes for disease and treatment method are available. And more detailed each patient’s data is not available on database. We authors mentioned these limitations on the discussion session.

Again we are really grateful for your considerations on our manuscript. And hope our work can be published on this honorable journal.

Reviewer 2 Report

Thank you for addressing my first round comments.

Author Response

  • We authors are really thank the reviewer for the thoughtful advice on our work.

And hope our work can be published on this honorable journal.

This manuscript is a resubmission of an earlier submission. The following is a list of the peer review reports and author responses from that submission.

Round 1

Reviewer 1 Report

The manuscript entitled “Does amputation negatively influence the incidence of depression in diabetic foot patients? addresses a very interesting topic for the clinical knowledge of health professionals , and has approach for publication in Applied Sciences. But there are some important methodological issues that make it unsuitable for publication. And in my consideration these are aspects that make it be considered that it is not current enough.

- I am mainly based on the fact that this work is not current, it was developed between the years 2011 and 2016, in addition, the references that the authors use can be considered not current, those prior to 2010 predominate.

- The methodology does not reflect in detail important aspects such as; it does not reflect how the depression risk assessment was done, how and where the data were recruited, recruitment of subjects, and data processing.

- Based on the aspects that I expose in point one and the data presented by the authors in the discussion, the results of the work do not contribute news and are also compared with non-current results.    

Author Response

Reviewer 1.

The manuscript entitled “Does amputation negatively influence the incidence of depression in diabetic foot patients? addresses a very interesting topic for the clinical knowledge of health professionals, and has approach for publication in Applied Sciences. But there are some important methodological issues that make it unsuitable for publication. And in my consideration these are aspects that make it be considered that it is not current enough.

- I am mainly based on the fact that this work is not current, it was developed between the years 2011 and 2016, in addition, the references that the authors use can be considered not current, those prior to 2010 predominate.

Response: We appreciate your comment. We totally agree with your opinion. I think this is a limitation of big data research using National Health Insurance Service data. I already mentioned limitation 1, but I will mention it more specifically and add the following

“Third, there is a limitation that data after 2016 are not included. Therefore, if access to HIRA data after 2016 is available,further analysis on this requires.”

- The methodology does not reflect in detail important aspects such as; it does not reflect how the depression risk assessment was done, how and where the data were recruited, recruitment of subjects, and data processing.

Response: Thanks for your kind reply. The depression risk assessment of this study was done by Charlson comorbidity index after propensity score matching as mentioned in section 2.2 Statistical analysis. Data and subject recruitment are presented in '2.1 Study design and Patient' section and 'Table 1'. The results were derived by substituting the diagnosis and treatment codes specified in Table 1 in the HIRA database.

- Based on the aspects that I expose in point one and the data presented by the authors in the discussion, the results of the work do not contribute news and are also compared with non-current results. 

Response: We appreciate your comment. We totally agree with your opinion. Again we authors think this is the limitation of big data research using National Health Insurance Service data. If the research system of National Health Insurance Service data develops further in the future, we will proceed with research that can reflect more recent trends using recent data. We authors are really grateful for your kind reply and hope our manuscript can be published in this honorable journal.

Reviewer 2.

-I have not observed any deficiencies in the article.

Response: We authors are really grateful for your kind reply and hope our manuscript can be published in this honorable journal.

Reviewer 3.

-Thank you for an interesting article. I suggest that you improve your Introduction and discuss the prevalence of diabetes and diabetic foot ulcers in Korea. In general, your literature review would need to be updated. There was wealth of studies published on depression in people with diabetes mellitus/diabetic foot ulcers and amputated limbs. Lines 48-56 – citations should be provided to support your claims.

Response: We appreciate your comment. We authors do agree that we lack reference on our statement in Introduction section so we put additional references on that part.

-I also suggest that you justify the selected statistical methods.

Response: Thanks for your kind reply. The depression risk assessment of this study was done by Charlson comorbidity index after propensity score matching as mentioned in section 2.2 Statistical analysis. Data and subject recruitment are presented in '2.1 Study design and Patient' section and 'Table 1'. The results were derived by substituting the diagnosis and treatment codes specified in Table 1 in the HIRA database.

-In Results, I would normally expect a table with statistical information of the included cases.

Response: We appreciate your comment. In Table 2, We included every variable with statistical information such as Hazard ration, Confidence interval and P-value.

-I found some of your statements confusing. You stated that “The incidence of depression was lower in the limb-saving group”. However, Figure 2 indicates the opposite.

Response: We appreciate your comment. There seems to have been a mistake in the writing process. We will change it as follows. “The incidence of depression was lower in the amputation group”

-In Table 2, what does ‘Reference’ mean?

Response: We appreciate your comment. This means that male for sex, amputation group for treatment method, and major amputation for amputation type are reference values for comparison with female, limb saving group and minor amputation.

-Ref: “With regard to risk factors contributing 134 to the risk of depression development, sex, number of comorbid diseases, and chronic 135 limb-saving procedure were found to be significant factors.” You have not presented this in your Results section.

Response: We appreciate your kindly comment. Section 3.2 ‘Contributing factors’ contains informations of the Results.

‘female sex was associated with a significantly higher risk of depression development than male sex (HR, 1.275; 95% CI, 1.117–1.451; P < 0.001). In the analyses, performed based on the comorbid diseases, the risk of depression increased with the number of underlying diseases; the effect was statistically significant (HR, 1.125; 95% CI, 1.098–1.151; P < 0.001). Of the treatment methods, the limb-saving method was associated with three times the risk of depression development compared to amputation (HR, 2.805; 95% CI, 2.478–3.183; P < 0.001).’

Discussion:

-I suggest that you compare your findings with the latest studies on the topic. How can you relate the following statement to your study: “They are also more prone to sex-specific conditions such as pregnancy, menstrual cycle changes, and post- partum stress”?

Response: We appreciate your comment. we mentioned the sentence “These factors may have a strong effect on a woman’s feeling of endangerment in terms of foot ulcer and its associated complications” on our manuscript to relate the previous finding to our study.

-More limitations could be identified and discussed.

Your conclusion could be better formulated. Please note that health professionals need to follow clinical practice guidelines in selection of optimal treatment methods for diabetic ulcer. Limb amputation is only necessary for the unsalvageable diabetic foot regardless of patients’ depressive symptoms. Health professionals should pay more attention to mental health of their patients in both, amputation and the limb-saving groups. Further studies could be suggested.

Response: We appreciate your comment. So, we change and add following sentences in conclusion part line 196-200. “Health professionals need to follow clinical practice guidelines in selection of optimal treatment methods for diabetic ulcer. Limb amputation is only necessary for the unsalvageable diabetic foot regardless of patients’ depressive symptoms. Health professionals should pay more attention to mental health of their patients in both, amputation and the limb-saving groups. Further studies could be suggested.”

Reviewer 2 Report

I have not observed any deficiencies in the article.

Author Response

and hope our manuscript can be published in this honorable journal.

Reviewer 2.

-I have not observed any deficiencies in the article.

Response: We authors are really grateful for your kind reply and hope our manuscript can be published in this honorable journal.

Reviewer 3 Report

Thank you for an interesting article. I suggest that you improve your Introduction and discuss the prevalence of diabetes and diabetic foot ulcers in Korea. In general, your literature review would need to be updated. There was wealth of studies published on depression in people with diabetes mellitus/diabetic foot ulcers and amputated limbs. Lines 48-56 – citations should be provided to support your claims.

I also suggest that you justify the selected statistical methods.

In Results, I would normally expect a table with statistical information of the included cases.

I found some of your statements confusing. You stated that “The incidence of depression was lower in the limb-saving group”. However, Figure 2 indicates the opposite.

In Table 2, what does ‘Reference’ mean?

Ref: “With regard to risk factors contributing 134 to the risk of depression development, sex, number of comorbid diseases, and chronic 135 limb-saving procedure were found to be significant factors.” You have not presented this in your Results section.

Discussion:

I suggest that you compare your findings with the latest studies on the topic. How can you relate the following statement to your study: “They are also more prone to sex-specific conditions such as pregnancy, menstrual cycle changes, and post- partum stress”?

More limitations could be identified and discussed.

Your conclusion could be better formulated. Please note that health professionals need to follow clinical practice guidelines in selection of optimal treatment methods for diabetic ulcer. Limb amputation is only necessary for the unsalvageable diabetic foot regardless of patients’ depressive symptoms. Health professionals should pay more attention to mental health of their patients in both, amputation and the limb-saving groups. Further studies could be suggested.

Author Response

Reviewer 3.

-Thank you for an interesting article. I suggest that you improve your Introduction and discuss the prevalence of diabetes and diabetic foot ulcers in Korea. In general, your literature review would need to be updated. There was wealth of studies published on depression in people with diabetes mellitus/diabetic foot ulcers and amputated limbs. Lines 48-56 – citations should be provided to support your claims.

Response: We appreciate your comment. We authors do agree that we lack reference on our statement in Introduction section so we put additional references on that part.

-I also suggest that you justify the selected statistical methods.

Response: Thanks for your kind reply. The depression risk assessment of this study was done by Charlson comorbidity index after propensity score matching as mentioned in section 2.2 Statistical analysis. Data and subject recruitment are presented in '2.1 Study design and Patient' section and 'Table 1'. The results were derived by substituting the diagnosis and treatment codes specified in Table 1 in the HIRA database.

-In Results, I would normally expect a table with statistical information of the included cases.

Response: We appreciate your comment. In Table 2, We included every variable with statistical information such as Hazard ration, Confidence interval and P-value.

-I found some of your statements confusing. You stated that “The incidence of depression was lower in the limb-saving group”. However, Figure 2 indicates the opposite.

Response: We appreciate your comment. There seems to have been a mistake in the writing process. We will change it as follows. “The incidence of depression was lower in the amputation group”

-In Table 2, what does ‘Reference’ mean?

Response: We appreciate your comment. This means that male for sex, amputation group for treatment method, and major amputation for amputation type are reference values for comparison with female, limb saving group and minor amputation.

-Ref: “With regard to risk factors contributing 134 to the risk of depression development, sex, number of comorbid diseases, and chronic 135 limb-saving procedure were found to be significant factors.” You have not presented this in your Results section.

Response: We appreciate your kindly comment. Section 3.2 ‘Contributing factors’ contains informations of the Results.

‘female sex was associated with a significantly higher risk of depression development than male sex (HR, 1.275; 95% CI, 1.117–1.451; P < 0.001). In the analyses, performed based on the comorbid diseases, the risk of depression increased with the number of underlying diseases; the effect was statistically significant (HR, 1.125; 95% CI, 1.098–1.151; P < 0.001). Of the treatment methods, the limb-saving method was associated with three times the risk of depression development compared to amputation (HR, 2.805; 95% CI, 2.478–3.183; P < 0.001).’

Discussion:

-I suggest that you compare your findings with the latest studies on the topic. How can you relate the following statement to your study: “They are also more prone to sex-specific conditions such as pregnancy, menstrual cycle changes, and post- partum stress”?

Response: We appreciate your comment. we mentioned the sentence “These factors may have a strong effect on a woman’s feeling of endangerment in terms of foot ulcer and its associated complications” on our manuscript to relate the previous finding to our study.

-More limitations could be identified and discussed.

Response: We appreciate your comment. We put additional sentence about our limitation on Discussion sentence.

Your conclusion could be better formulated. Please note that health professionals need to follow clinical practice guidelines in selection of optimal treatment methods for diabetic ulcer. Limb amputation is only necessary for the unsalvageable diabetic foot regardless of patients’ depressive symptoms. Health professionals should pay more attention to mental health of their patients in both, amputation and the limb-saving groups. Further studies could be suggested.

Response: We appreciate your comment. So, we change and add following sentences in conclusion part line 196-200. “Health professionals need to follow clinical practice guidelines in selection of optimal treatment methods for diabetic ulcer. Limb amputation is only necessary for the unsalvageable diabetic foot regardless of patients’ depressive symptoms. Health professionals should pay more attention to mental health of their patients in both, amputation and the limb-saving groups. Further studies could be suggested.”

Reviewer 4 Report

A significant criticism in the present study is that were included only a very few data.

The amputation itself may not cause depression in the absence of significant problems after amputation such as infections, need for hospitalization, a second or third amputation, antiglycemic treatment (e.g., change from pills to injections), development of neuropathic pain, and development of other complications. Unfortunately, all these data are missing.

Moreover, the authors reported that ‘’ no significant difference was identified between the minor amputation and major amputation groups’’.  Is there an explanation for that?  Because major amputations cause a more severe problem.

Another significant issue is the following paragraph:’’ Limb-saving treatment was associated with a higher depression than amputation. With regard to risk factors contributing to the risk of depression development, sex, number of comorbid diseases, and chronic limb-saving procedure were found to be significant factors’’.  But data about those factors are missing, and that has not been reported the type of amputation.

Otherwise, the explanation is inadequate because limb-saving treatment is expected to be associated with a lower incidence of depression.

I think that the methodology is not appropriate. The manuscript is generally clearly written, but the discussion/conclusions are not acceptable.

   Overall, the data could be of interest if the authors add the data mentioned above.  Otherwise, the manuscript cannot be published.

Reviewer 5 Report

A very interesting paper and topic definetly worth publishing.

However some more detail is required in the manuscript especially in the methods section.

More detail/defentions for codes for diabetic foot, depressive disorder and treatment methods employed in the paper and not in supplementary material. Not everybody is familiar with the codes you include so you need to define what each code includes. clearer inclusion/exclusion criteria.

Change term 'sex' to gender please.

Reviewer 6 Report

1. The study presents the results of original research.

2. Results reported have not been published elsewhere.

3. Experiments, statistics, and other analyses are performed to a high technical standard and are described in sufficient detail.

4. Conclusions are presented in an appropriate fashion and are supported by the data.

5. The article is presented in an intelligible fashion and is written in standard English.

6. The research meets all applicable standards for the ethics of experimentation and research integrity.

7. The article adheres to appropriate reporting guidelines and community standards for data availability.

Reviewer 7 Report

Congratulations on the job. I think it is very interesting and highlights the importance of approaching these patients from a psychological point of view. For its publication it is necessary that you make some changes, you must include more recent references in the introduction and discussion; as well as completing the introduction.

Round 2

Reviewer 1 Report

Thank the authors for the work they did to improve the manuscript. But they forgot to mark the modifications made. Despite this, I still consider that the work does not offer a present time to consider its publication.

We can see a reflection of this that the authors discuss with results published after 2016 (data collections). We can observe that the authors discuss their results with works published after 2016, referring to previous studies.

Line 143: ref 3. (2019)

Line 153: ref. 8 and 20. (2018 and 2016)

Line 167: ref 24 (2017)

Reviewer 3 Report

Dear Authors

Thank you for your revisions. I feel that more work is needed. It appears that my comments were either taken lightly or not addressed at all. I suggest that you spend more time on revisions, and address my comments and suggestions.

Kind regards,